# Early Life Stress, Brain Development, and Obesity Risk: Is Oxytocin the Missing Link?

**DOI:** 10.3390/cells11040623

**Published:** 2022-02-11

**Authors:** Georgia Colleluori, Chiara Galli, Ilenia Severi, Jessica Perugini, Antonio Giordano

**Affiliations:** 1Department of Experimental and Clinical Medicine, Marche Polytechnic University, Via Tronto 10/A, 60020 Ancona, Italy; g.colleluori@pm.univpm.it (G.C.); c.galli@pm.univpm.it (C.G.); i.severi@staff.univpm.it (I.S.); j.perugini@staff.univpm.it (J.P.); 2Center of Obesity, Marche Polytechnic University-United Hospitals, 60020 Ancona, Italy

**Keywords:** hypothalamus, energy homeostasis, postnatal development, lactation, leptin, trauma, eating disorders, dysfunctional eating

## Abstract

Obesity disease results from a dysfunctional modulation of the energy balance whose master regulator is the central nervous system. The neural circuitries involved in such function complete their maturation during early postnatal periods, when the brain is highly plastic and profoundly influenced by the environment. This phenomenon is considered as an evolutionary strategy, whereby metabolic functions are adjusted to environmental cues, such as food availability and maternal care. In this timeframe, adverse stimuli may program the body metabolism to maximize energy storage abilities to cope with hostile conditions. Consistently, the prevalence of obesity is higher among individuals who experienced early life stress (ELS). Oxytocin, a hypothalamic neurohormone, regulates the energy balance and modulates social, emotional, and eating behaviors, exerting both central and peripheral actions. Oxytocin closely cooperates with leptin in regulating energy homeostasis. Both oxytocin and leptin impact the neurodevelopment during critical periods and are affected by ELS and obesity. In this review article, we report evidence from the literature describing the effect of postnatal ELS (specifically, disorganized/inconstant maternal care) on the vulnerability to obesity with a focus on the role of oxytocin. We emphasize the existing research gaps and highlight promising directions worthy of exploration. Based on the available data, alterations in the oxytocin system may in part mediate the ELS-induced susceptibility to obesity.

## 1. Obesity: Epidemiology, Etiopathophysiology, and Early Development

Obesity, defined as a body mass index (BMI) ≥30.0 kg/m^2^, is a multifactorial, chronic, and relapsing disease that has spread to pandemic proportions during the last decades [1,2]. Obesity prevalence has in fact nearly tripled since 1975 and its incidence is expected to increase further in the near future [1,2,3]. This estimate is not surprising as the incidence of obesity among children is rising steeply [4] and the condition is usually maintained throughout life [5]. Notably, in 2016, more than 1.9 billion people (39% of adults) worldwide were overweight (25.0 ≤ BMI < 30.0 kg/m^2^) and more than 650 million suffered from obesity (13%). Furthermore, in 2020, 39 million children under the age of 5 were overweight or suffered from the disease [6]. Obesity is associated with a higher risk of developing over 200 medical complications, including insulin resistance, type 2 diabetes mellitus, hypertension, metabolic syndrome, cardiovascular disease, and several types of cancer. For the above reasons, obesity is recognized as the fifth leading cause of death worldwide and as a major burden for the global healthcare systems [6,7,8].

From a biological standpoint, obesity results from the inability to ensure energy homeostasis, an impairment referred to as energy balance dysfunction. This concept is often simplistically ascribed to excessive energy intake (eating) and low energy expenditure (physical activity), hence to an “unhealthy lifestyle”. Nonetheless, the etiology of obesity is complex and multifactorial [9]. According to the Foresight Study, multiple environmental (i.e., food industry, pollution, education, culture, access to healthcare), psychological (individual and social), and biological (genetic, epigenetic, endocrinological) factors not only contribute to determine the obesity risk, but also positively and negatively influence each other in triggering the disease and its morbidity [9]. Eventually, the interplay between these causal factors results in a dysfunctional regulation of the energy balance, hence abnormal energy intake and expenditure.

The pathophysiology of obesity involves multiple organs. The chronic positive energy balance results in an excessive accumulation of lipids within the adipose depots and in different cytotypes. This phenomenon is responsible for adipose tissue expansion, inflammation, and lipotoxicity and deeply compromises several organ functions [7,10,11,12,13]. For example, obesity-related lipotoxicity and chronic inflammation may result in non-alcoholic fatty liver steatohepatitis; skeletal muscle dysfunction, i.e., sarcopenic obesity [12,14]; and pancreatic β-cell impairment [15]. Overnutrition and obesity are associated with inflammation and endoplasmic reticulum stress in the mediobasal hypothalamus, which hosts the centers regulating the energy balance [16]. Organ dysfunction in turn impairs whole-body energy homeostatic abilities and triggers a vicious cycle that underpins the chronic and relapsing nature of obesity [1,7].

The energy balance is controlled by the central nervous system (CNS) [17,18,19,20]. The master regulator is the hypothalamus, where all signals from other brain areas and from the periphery are integrated and translated into specific behavioral, autonomic, and endocrine outputs [17,18,19,21,22]. The crucial role of the CNS in obesity susceptibility is documented by recent genome-wide association studies that implicated pathways related to synaptic function, extracellular matrix composition, glutamate signaling [23], and brain G protein-coupled receptors as key factors governing BMI variations [24].

The maturation of the central neural circuitries involved in energy balance control is not completed at birth but also occurs during early postnatal life. In mammals, postnatal ages are denoted by critical developmental periods during which organs and neural systems are highly plastic. In this timeframe, adverse nutritional, social, and environmental cues may program body metabolism to maximize energy accrual to cope with hostile conditions. Accordingly, the prevalence of obesity is higher among individuals exposed to early life stress (ELS) during both the pre- and postnatal periods [25].

In rodents, thermogenic brown adipose tissue (BAT) is present at birth to sustain pups’ survival, whereas white adipose tissue (WAT) progressively develops during the first postnatal weeks. This timing may reflect an evolutionary strategy aimed at adjusting metabolic functions to environmental cues, such as maternal care and food availability. Obesity disease is hence rarely due to environmental and/or genetic factors alone and results from the interaction between the individual’s biological characteristics and the environment in which he lives in [1]. For this reason, obesity is recognized as a preventable disease [1].

The study of the cellular and molecular mechanisms at the basis of the postnatal differentiation of the energy homeostatic circuitry may shed light on the etiopathogenetic basis of several forms of obesity and may offer new targets for intervention and prevention. Given the strong association between disorganized/inconstant maternal care and obesity risk [26,27], the identification of the biological pathways and mediators of such a link is of relevance. The oxytocin system is a highly promising candidate, given its role in maternal bonding, response to stress, and feeding behavior [28,29,30]. In this review, we report evidence from the literature documenting the effect of ELS (specifically postnatal stress induced by disorganized/inconstant maternal care) on obesity vulnerability, with a particular focus on oxytocin (Oxt) and leptin (Lep) roles in rodent models. We emphasize the existing gaps in the literature and highlight promising research directions worthy of exploration.

## 2. Early Life Stress

Clinical observations and studies by authoritative psychiatrists and pediatricians, such as John Bowlby and Donald Winnicott, emphasized the crucial role of early life experiences, in particular the relationship between the infant and the mother (or caregiver), in shaping individuals’ psychological health [31,32]. Specifically, a disorganized care attitude toward the offspring results in abnormal attachment/bonding to the parental figure and in the development of dysfunctional behaviors (e.g., depression, drug addiction, and eating disorders) in adulthood [26,27,30,31]. During early life, parental–offspring interactions exert a critical influence on offspring’s developmental trajectories [33]. In animal models, ELS is usually reproduced by maternal separation (MS), which consists in separating pups from their dam for 180 min per day, or by limiting the nesting and bedding material (LN), a procedure that results in disorganized maternal care and hence stress among pups [34,35]. Maternal care anomalies lead to abnormal developmental courses underlined by very complex and poorly characterized neuroendocrine cascades [36]. Growth in an unpredictable environment, where maternal care is not consistent, may result in a neurometabolic programming aimed at maximizing energy accrual and minimizing its waste, a possible adapting strategy to face hostile conditions. As a result, the rewarding value of feeding may also be abnormally enhanced [21].

In adult humans and experimental animals, acute stress suppresses appetite while chronic stress often leads to weight gain due to the persistent overactivation of the hypothalamic-pituitary-adrenal (HPA) axis and to opioid release; both systems promote “comfort eating” as a coping strategy [18]. Although the effects of acute and chronic stress on eating behavior and metabolic health have been extensively studied in adults, the mechanisms through which ELS eventually affects them in adulthood are unclear [26] (Table 1). Exposure to stress during early life is associated with overweight and obesity in adult humans [26,27]. ELS is known to negatively impact neurobiological, cognitive, social-emotional, behavioral, and physical development in humans and animal models. In the timeframe characterizing early development, known as the critical period, the infants’ brain is extremely plastic. Intense phenomena of pruning and synaptogenesis are necessary to shape neural circuit and to complete brain maturation, which concludes only after adolescence [33]. In this developmental process, a crucial role is also played by glial cells, which actively modulate synaptic formation, pruning, neurovascular coupling, and phagocytosis [37]. Glial cells also act as metabolic sensors, integrating a multitude of signals to adapt to the environment [38]. This period of high plasticity is deeply influenced by life experiences and is characterized by a marked vulnerability to negative stimuli, which can shape the neural network [33,36]. However, relatively little is known about how these critical periods impact the development of peripheral organs (e.g., adipose tissue) and their crosstalk with the CNS. The study of obesity etiopathogenesis requires not only a longitudinal approach that explores in depth early life events, but also a comprehensive research strategy investigating the crosstalk between the CNS and peripheral organs. This notion can be summarized as the study of the development and function of the neuroendocrinology of energy balance, also defined as neurometabolic programming in this article.

Although the biological effects of ELS on cognitive function, anxiety, and depression-like behavior have been widely investigated [39,40,41], their impacts on central regulation of the energy balance, eating behavior, adipose tissue development, and metabolic health have not been explored in depth and only a handful of studies, with very heterogeneous experimental designs, are available on the topic [26,39,42,43,44,45,46,47].

## 3. Impact of ELS on Metabolic Health and Eating Behavior: Evidence from Rodent Studies

Studies conducted in humans and animal models established that ELS negatively affects cognitive, social-emotional, behavioral, and physical development [48,49]. However, animal studies investigating the effect of ELS on body weight in adulthood report conflicting results, with some describing initial weight loss and weight normalization with advancing age [40,47], and others reporting weight gain [39,50,51] or no variations [45]. Importantly, most studies involved rats and reported conflicting evidence regarding the consequences of ELS followed by an obesogenic environment in adulthood. For example, Paternain and colleagues studied the metabolic parameters of female rats exposed to daily MS (180 min) from postnatal day (PND) 2 to 21 followed by 35 days of a high-fat sucrose diet (HFSD) at 2 months of age [42]. The combination of MS and HFSD resulted in a lower weight of the subcutaneous and periovarian WAT depots compared to HFSD alone. On the other hand, the HOMA index (an indicator of insulin resistance) was significantly higher in the former rats, even though the two groups did not show significantly different food intake. In contrast, male rats exposed to ELS (LN, PND 2–21) followed by a HFSD failed to gain weight and showed normal glucose control in adulthood [43]. In a mouse study, Yam and co-workers investigated the effects of LN (PND 2–9) followed by a moderate Western diet in adulthood (PND 42–98) and found that male mice displayed a lower relative fat mass content but a higher increase in the fat percent in response to the Western diet [44]. In contrast, Eller and colleagues reported that MS (PND 1–21) coupled with a HFSD (from age 16 weeks) resulted in a higher body weight and fat mass in C56BL/6 male mice only when they had free access to a treadmill [46]. The authors attributed this finding to reduced voluntary exercise, hence lower energy expenditure [46]. Another study found that MS and early weaning induced an increase in fat mass (but not in body weight) in male and female mice [52]. When HFD was added to the stress protocol, both males and females became hypertensive compared to controls, but only females showed a persistently higher fat mass, adipocyte hypertrophy, and impaired metabolic outcomes, such as dyslipidemia and hyperinsulinemia.

Interestingly, the eating behavior of animals receiving a chow diet did not seem to be affected by ELS. However, a recent investigation demonstrated that male and female rats subjected to MS (PND 2–14) ate more highly palatable food than controls while their weight and chow diet consumption were unchanged [47]. These data suggest that ELS may impact homeostatic and hedonic feeding in different ways.

In conclusion, investigation of the effects of ELS on adult metabolic health and eating behavior in animal models has highlighted gaps and inconsistencies that are not only due to the complexity of the topic, but also to the diverse protocols applied. Sex-specific differences in the response to ELS and dietary interventions may play a role in such variability. According to recent studies, ELS impacts brain development in a sex-specific manner, a phenomenon that affects several behavioral and metabolic features [53,54]. Such disparity may underlie the different adipose tissue expansion and distribution characterizing males and females [55,56,57,58] and the diverse sex-dependent susceptibility to obesity-related comorbidities [55].

The study and the interpretation of the available data on the topic is particularly challenging due to (1) the lack of uniformity in the ELS and dietary protocols; (2) the differences in the response to ELS based on sex, age, and species; and (3) the disparities in the outcomes chosen for the analyses.

## 4. Oxytocin: The Neuroendocrine Hub of Social Bonding, Stress, Eating Behavior, and Metabolic Health

The neurohormone Oxt is closely involved in the effects exerted by abnormal infant care. Oxt is produced by magnocellular and parvocellular neurons located in the hypothalamic paraventricular (PVN) and supraoptic (SON) nuclei (Figure 1A–E) and plays a pivotal role in the regulation of a variety of behaviors, including social, emotional, sexual, eating, and addiction behaviors [21]. Magnocellular neurons are found in the PVN and SON and mainly project to the neurohypophysis, where Oxt is released into the circulation; parvocellular neurons are mainly located in the PVN but are also scattered in other hypothalamic and extrahypothalamic areas and project to different hindbrain regions, such as the nucleus of the solitary tract (NTS) [59,60]. Interestingly, PVN and SON magnocellular Oxt neurons develop axon collaterals projecting to forebrain limbic regions (e.g., prefrontal cortex, nucleus accumbens, anterior and central amygdala, bed nucleus of the stria terminalis (BNST), hippocampus) (Figure 2A–F). This finding has only been described in advanced vertebrates and is believed to have developed together with the social and emotional behavioral complexity of species.

Interestingly, since their axons and/or dendrites are also found in the proximity of the third ventricle and among ependymal cells in direct contact with the cerebrospinal fluid (Figure 1D,E), Oxt neurons may directly release the nonapeptide into the third ventricle and/or act as biochemical sensors. Importantly, Oxt is also produced by peripheral tissues (e.g., corpus luteum, uterus, adrenal cortex, testis), and its receptor (Oxtr) shows a wide peripheral distribution [60]. For example, Oxt is expressed by the nerves of myenteric and submucous ganglia along the gastrointestinal tract [61,62], where it modulates gastric motility and inflammation [63]. It also acts on the peripheral organs involved in metabolic homeostasis, such as adipose tissue, liver, and pancreas [64]. The peripheral metabolic action of Oxt is discussed in Section 7.

Oxtr has been detected in various brain areas, including those where Oxt neuron projections are not documented [60]. Such mismatch, together with the Oxt detection in the cerebrospinal fluid, suggests that cerebral Oxt signaling involves synaptic and volume transmission [65]. Dendritic Oxt release occurs independently of neuron spiking activity, and allows Oxt action on different target cells, even far from the area of secretion [66]. In this context, it should be stressed that since the regulation of Oxtr expression is mediated by sex steroids, sex dimorphism should always be considered when investigating the Oxt system [60]. Compared to males, female mice display a higher number of Oxt-immunoreactive neurons in the PVN and SON and more Oxt projections to several brain mesolimbic areas [67]. However, a study of Oxt^Cre^;Z/AP mice, where Oxt neurons are labeled with alkaline phosphatase, documented similar Oxt projections in both sexes [68]. Another possible confounder is the close homology between the Oxt and the vasopressin systems, especially in the context of ELS [60]. Since this topic is beyond the scope of this review, the reader is referred to [28,69,70,71] for further details.

Oxt stimulates maternal care, maternal–infant attachment, and social bonding and can attenuate the response to stress, anxiety, and depression [21,28]. It also has a crucial role in the regulation of the stress system by reducing HPA activity and by supporting the parasympathetic nervous system [28,72,73]. Recent works on the neurobiological basis of attachment, coupled with studies on children adopted from orphanages, suggest that there may be a sensitive period for the development of Oxt–dopamine connections (particularly in the nucleus accumbes of the striatum), which exerts enduring effects on the neurobiology of social relationships [72], for instance, by strengthening the ability to buffer stress [73].

Interestingly, Oxt also attenuates addictive behaviors and inhibits appetite [72]. Consistent with these data, there is a mounting body of evidence pointing at Oxt role in promoting weight loss and ameliorating obesity-related metabolic dysfunctions [29,64,74,75]. Intranasal Oxt administration is currently being tested for the treatment of obesity as this route facilitates an increase in the central concentrations of the nonapeptide through channels surrounding the trigeminal and olfactory nerve fibers [76,77,78]. Considering the complex and multiple functions of Oxt, which affects aspects as diverse as mother–infant bonding, eating behavior, and stress response, its potential role in determining the impact of ELS on eating behavior and metabolic health deserves further investigation. The existing evidence linking Oxt to ELS, eating behavior, and metabolic health is reviewed below.

## 5. Oxytocin System: Early Development and Impact of ELS

Rodent studies have demonstrated that early adverse experiences may have a strong impact on the way the Oxt system is shaped. Oxt neurons progressively increase from PND 2 to 21, reaching maturation by the second postnatal week [70]. In contrast, Oxt axons reach their targets only in early adulthood, when Oxtr is already widely expressed in various regions. Specifically, Oxtr is detected in females from embryonic day 14 and in males from PND 2 [60]. Importantly, Oxt-synthesizing neurons play a fundamental role in their own maturation: local Oxt release activates Oxtr expressed on Oxt neurons, resulting in further Oxt discharge. These data explain how deeply early variations in Oxt levels can impact the maturation of Oxt neurons [21,66,70]. It is hence possible that postnatal environmental stimuli affect Oxt production and the development of the Oxt system, resulting in the exclusive retention of functional pathways necessary to face external conditions. This would also explain the interindividual variability of Oxt neuron projections revealed by Oxt system mapping [68]. Notably, the different spatio-temporal Oxtr expression patterns described in males and females during development suggest a distinct and sex-specific sensitivity to Oxt levels during this critical period [60].

Altered Oxtr expression and binding in response to changes in maternal care were described in several brain regions in different animal models [36,79]. Maternal high licking and grooming (LG) result in increased Oxt expression at PND13 while low LG leads to reduced Oxtr protein levels and receptor binding in several females’ brain regions (e.g., PVN, central nucleus of the amygdala) [36,69,79]. Furthermore, while one study detected a higher number of Oxt-positive cells in the PVN of adult male mice exposed to MS [80], other investigations with a similar protocol reported a lower number of these cells in the SON and PVN [71,73,81]. In addition, higher Oxtr expression and a higher number of Oxt projections to the basolateral amygdala were described in male mice exposed to MS [81]. Early adversity usually results in reduced circulating Oxt [73]. However, Oxt levels may actually rise in response to prolonged exposure to adverse stimuli, possibly to protect the system from the harmful effects of stress [73]. Importantly, Oxt detection assays are often unreliable and several studies dosing Oxt in circulation or in cerebrospinal fluid have been judged erratic [60].

MS influences Oxtr binding and expression in different extrahypothalamic brain regions [30] and modulates dopaminergic, neuroendocrine, stress, and immune response system function [73]. These biological features may underpin the association between ELS, impaired attachment, the Oxt system, and addiction [73]. A recent meta-analysis documented lower plasma Oxt and a reduced or negative response to intranasal Oxt administration among individuals who experienced childhood adversity [82]. In addition, an insecure attachment style was associated with lower Oxtr expression in peripheral blood mononuclear cells of women [83]. In summary, early experience shapes the Oxt system and manipulation of maternal care from infancy induces lasting changes in Oxtr expression, whose underlying mechanisms and implications have not been fully elucidated.

## 6. Leptin, Brain Development, and Early Life Stress: Beyond Energy Homeostasis Regulation

Lep is an important mediator of brain maturation that is capable of shaping the hypothalamic neural circuits involved in energy balance regulation [84]. It is an adipokine that acts as a signal of the body energy stores, whose production and release into the circulation increase with white adipocyte hypertrophy [85,86,87]. It has a central anorexigenic role that is mainly exerted through action on the hypothalamic arcuate nucleus (ARC), where it inhibits orexigenic neuropeptide Y-Agouti-related protein (NPY-AgRP) neurons and activates anorexigenic pro-opiomelanocortin (POMC) neurons, two types of primary order neurons (Figure 3A–C) [86]. These two neuronal populations project to other hypothalamic nuclei (secondary order neurons)—the ventromedial (VMH), dorsomedial (DMH, Figure 3D), and lateral hypothalamic (LH) nuclei—that are involved in energy balance regulation and are also direct Lep targets [86]. Importantly, hyperleptinemia and leptin resistance are common in obesity disease [86]. Lep is detected in the circulation during early life, even though WAT achieves maturation during the first postnatal weeks [88]. In early life, Lep can cross the blood–brain barrier, reaches its receptor (Lepr), which is widely distributed in several brain areas, and exerts a trophic action on the developing brain. The brain of *ob*/*ob* and *db*/*db* mice, lacking Lep and Lepr, respectively, displays lower cell density, altered hypothalamic dendritic orientation, and immature glial and synaptic protein expression patterns [88,89,90]. The finding that in *ob*/*ob* mice, such alterations could be normalized with Lep injection in early life but not in adulthood supports the key role of Lep during the critical neurodevelopment periods [90,91,92,93]. Notably, several hypothalamic nuclei progressively acquire Lep responsiveness (p-STAT3 activation) in the first postnatal days (up to PND 15) when adipokine does not regulate the energy balance yet [84]. This suggests that during this developmental stage, Lep actually exerts a neurotrophic role [84]. Deficient Lep signaling in early life results in an altered number of axonal projections from the ARC (AgRP and POMC neurons) to the PVN [93,94,95]. In *ob*/*ob* mice, this alteration can be restored only by Lep administration on PND 4 to 12 [93]. In a model of leptin receptor (Lepr) deficiency, Romos-Lobo and colleagues documented a similar impairment, which could, however, be rescued even when the induction of *Lepr* expression occurred in adulthood [94]. Nevertheless, the same group reported persistent anomalies in the hypothalamic expression of several regulators of energy metabolism (e.g., *Pomc*, *Cart*), in brain weight, and metabolic outcomes, such as energy expenditure and insulin sensitivity.

Based on these findings, environmental stimuli affecting Lep levels in early life may have an enduring effect on brain maturation. ELS (specifically MS) was reported to blunt hypothalamic Lep signaling and its anorectic effect in adults [96]. ELS (LN; PND 2–9) resulted in a short- (PND 9) and long- (PND 180) term reduction in plasma Lep and in its WAT expression in both sexes [44]; however, this study did not consider the contribution of the diet to Lep variations. Higher plasma Lep and increased Lep mRNA expression in gonadal, but not inguinal WAT, have been described in females exposed to ELS (MS and early weaning) plus HFD [97]. Interestingly, a recent meta-analysis found a reduction in circulating Lep due to acute stress [98].

PVN oxytocin neurons express Lepr. In fact, in adults, Lep action is partly mediated by Oxt, since administration of Lep and Oxt antagonists blunts the anorectic effects of Lep (further detailed in the next section) [99]. However, the relationship between the trophic action of Lep and the development of the oxytocinergic system in early life has not been investigated (Table 1). Since in early development, environmental stimuli, especially nutritional status and stress, can permanently shape neurometabolic programming, the investigation of the crosstalk between these hormone systems in early life is central to obesity research. Furthermore, the potential trophic role of Lep on Oxt neuron maturation has never been explored before.

## 7. Oxytocin, Eating Behavior, and Metabolic Health

Genetic Oxt ablation results in deep lactational impairments, which lastly lead to offspring death, whereas other biological functions regulated by Oxt (e.g., uterine contraction) are preserved. However, Oxt and Oxtr knockout (KO) models display late-onset obesity with a progressive increase in relative fat mass [100,101]. Oxtr KO animals show a higher mesenteric, perirenal, and epididymal WAT weight and a greater white adipocyte cross-sectional area, whereas their brown adipocytes display phenotypic features of white fat cells [100,101]. The same model does not exhibit increased food intake (chow diet) but has a lower core body temperature, reflecting impaired thermoregulation [101,102]. The resulting reduced energy expenditure may be responsible for the increased WAT mass [101,102]. While Oxt KO mice do not consume significantly more chow diet (homeostatic feeding), “hedonic feeding”, which is studied by evaluating sucrose and/or saccharine intake, significantly increases [103]. According to a recent study, Oxt intracerebroventricular (i.c.v.) administration inhibited sucrose consumption without affecting the post-ingestive reinforcing properties of palatable food [104]. Among the mechanisms that are the basis of this phenomenon is Oxt-mediated suppression of VTA phasic dopamine neuron activity in response to cues associated with palatable food [104] (Figure 4A,B). Consistent with these observations, patients with Prader–Willy syndrome, which is characterized by a reduced number of Oxt neurons in the PVN, suffer from hyperphagic obesity [29]. In rodents, Oxt administration (subcutaneous or intraperitoneal) reduces food intake and fat mass without affecting lean mass [100]. In contrast, due to the very heterogeneous methods used to assess circulating Oxt [60], it is unclear whether plasma Oxt is changed in humans with obesity, since some studies reported a reduction [105] and others an increase [29].

Interestingly, the tissues expressing the highest *Oxtr* levels in wild-type female mice are visceral (vWAT) and subcutaneous (sWAT) WAT followed by the uterus and ovary [100]. *Oxtr* expression is relatively lower in males, where it is most abundant in vWAT and the testis [100]. On the other hand, in interscapular brown adipose tissue (iBAT), *Oxtr* expression is relatively low in both sexes [100]. The effect of Oxt on adipocytes is a relatively new area of investigation. A study reported a negative relationship between Oxtr expression in the adipose tissue and circulating Oxt in obese Zucker rats [105]. Studies performed in vitro (on the 3T3-L1 cell line) show a progressive increase in the mRNA levels of *Oxtr* in differentiating adipocytes, paralleled by an increase in the expression of *Lep* and *PPARγ*, a master regulator of adipocyte differentiation [106]. The authors confirmed the same trend for increasing *Oxtr* expression with increasing adipose tissue weight during mice growth (7 vs. 14 weeks), a finding again accompanied by increasing *Lep* and *PPARγ* levels but not by a similar increase in the hypothalamic *Oxt* expression. Oxt treatment in vitro stimulates Lep release and adipocyte lipolysis, which, combined with the above findings, suggests that *Oxtr* increases with adipocyte differentiation and lipid accumulation to ensure lipid homeostasis [107]. According to this study, HFD increased *Oxtr* expression and the *Lep*, *PPARγ*, and adipocyte cross-sectional area in all adipose depots. Furthermore, fasting did not affect adipocyte *Oxtr* expression whereas refeeding resulted in an increase. All documented changes were coupled with *Lep* expression variations. However, a major limitation of this study is related to the assessment of *Oxtr* mRNA only [106]. On the other side, intraperitoneal Oxt injection resulted in reduced food intake, visceral fat mass, fatty liver, and improved glycemic control in obese mice [108]. Importantly, Oxt treatment restored the acute anorectic effect of Lep in obese leptin-resistant male mice [109]. However, chronic Oxt and Lep coadministration had no significant effect on body weight or food intake compared to the separate treatment with each peptide [109]. In ovariectomized mice, Oxt administration partly rescued the obese phenotype induced by the lack of estrogens and reduced fat mass and plasma Lep by inhibiting adipogenesis, especially in the intrabdominal depot [110]. OXT-mediated inhibition of adipogenesis is consistent with the results from a study that administered carbetocin (OXT analog) to human multipotent adipose-derived stem cells (hMADS) [111]. Deblon and colleagues confirmed the lipolytic effect of i.c.v. Oxt administration on epididymal WAT (eWAT) in adult mice [112]. In particular, increased adipocyte lipolysis and fatty acid oxidation were mediated by Oxt-induced oeylthanolamide production, which activates *PPARα*. The authors also found lower food intake among HFD Oxt-treated animals, which, compared to paired fed mice, showed greater lipid utilization, a lower eWAT content, and lower circulating Lep. On the other hand, Oxt-treated C56BL/6 mice (subcutaneous injection) increased plasma Lep compared to controls [113].

The action of Oxt on adipose tissue and its close connection with Lep secretion raise several questions regarding their interplay in energy balance regulation. *Ob*/*ob* mice treated with Oxt lose fat mass (especially eWAT) in a process mediated by both a reduction in food intake and in energy efficiency (increased futile lipid cycle in adipocytes) [113]. The anorexigenic Lep action is in fact partly mediated by Oxt. Accordingly, in fasted rats blockage of Oxt signaling attenuates the leptin-induced body weight reduction [99]. Specifically, i.c.v. Lep administration activates PVN-Oxt neurons projecting to the NTS in chow-fed and in DIO rats [99,114] (Figure 4C,D). Furthermore, mice lacking Socs3, the inhibitor of Lep signaling in the mediobasal hypothalamus (ARC, DMH, VMH, and LH), displayed reduced food intake, lower diet-induced weight gain, and a greater sensitivity to satiety signals, in a process mediated by the leptin-induced Oxt action on the NTS [115].

Oxt plays a well-established role during parturition and lactation, two periods when adipose tissues undergo profound changes [8]. Since Oxt is essential for lactation and exerts a lipolytic effect on fully differentiated white adipocytes, it is possible that such a neuropeptide mediates the energy redistribution, in particular lipids relocation from adipocytes to epithelial mammary gland cells, to support milk production.

Given that the changes in the body weight of Oxt-treated or Oxt KO animals are not fully explained by changes in WAT metabolic and endocrine functions, several studies have investigated the influence of Oxt on energy expenditure, particularly on the thermoregulation abilities [101,102,116,117]. Methods that activate BAT energy dissipation through heat production have crucial potential in treating obesity [118]. Cold exposure increased hypothalamic Oxt levels and Oxtr expression in iBAT and inguinal WAT. Such changes were paralleled by an increased expression of uncoupling protein 1, responsible for energy dissipation in the form of heat [116]. In line with these findings, Oxt treatment of brown C3H10T1/2 cells activated the thermogenic pathway and increased lipid utilization in a process dependent on PRDM16, a master regulator of brown adipocyte differentiation [116].

The above data highlight the critical importance of the interplay between the Oxt system, feeding behavior, and metabolic health. However, whether the impact of ELS on the Oxt system is responsible for the alterations in feeding behavior and metabolic health has not been investigated.

## 8. ELS, Oxytocin, Eating Behavior, and Metabolic Health: Future Research Directions

Recent studies documented the association between the alteration in the Oxt system induced by ELS and the development of cardiovascular diseases [119,120,121]. However, little is known about the link between Oxt, ELS, and metabolic health. While data on the relationship between ELS and Oxt and between ELS and metabolic health are scanty, evidence linking ELS to metabolic health, focusing on the Oxt system, is not documented in the literature. Specifically, it is unclear how ELS induces changes in the action of and sensitivity to Oxt and how these changes influence adipose tissue development, energy balance, metabolic health, and feeding behaviors in adulthood (Table 1). The sex-dependent differences in the Oxt system may affect the adipose proliferative niches (hence, adipose tissue development) and may partly contribute to the distinctive fat mass distribution and susceptibility to obesity comorbidities described in males and females. Furthermore, while the effect of Oxt manipulation in early development has been widely investigated [70], little is known about the function of endogenous Oxt at such a time [30]. Several lines of research have demonstrated how spatio-temporal Oxtr expression varies in response to environmental cues, especially during critical periods. It is widely accepted that Oxtr may be a developmental plasticity gene that acts as a transducer of the social environment to finetune the experience-dependent plasticity of the social brain [60]. Therefore, Oxtr expression and distribution, as opposed to circulating Oxt levels per se, may be of particular significance. As noted above, Oxt influences reward-related behaviors in different ways: while it reduces motivated behavior toward palatable food, it modulates social rewards and attention-orienting responses to external social cues [104]. It is reasonable to hypothesize that dysfunctional social attachment to the parental figure during early development (MS or LN) alters the oxytocinergic system maturation such that the reward response to feeding prevails on the reward response to social cues. This paradigmatic model may be employed not only to study obesity vulnerability, but also psychiatric diseases, such as eating disorders, depression, and addiction. Indeed, the limitations related to the use of animal models for the study of such complex diseases should be acknowledged. Growth in an unpredictable environment, where maternal care (feeding) is not constantly ensured, may result in a neurometabolic programming that maximizes energy accrual and minimizes its waste. As a result, besides the enhanced reward value of food, the ability to store energy to cope with an adverse environment may also be increased (Figure 5). Deeper investigation of the biological underpinnings of ELS-induced metabolic and eating behavior abnormalities is warranted to characterize unexplored mechanisms responsible for obesity and to identify novel preventive and/or therapeutic strategies.

## Figures and Tables

**Figure 1 cells-11-00623-f001:**
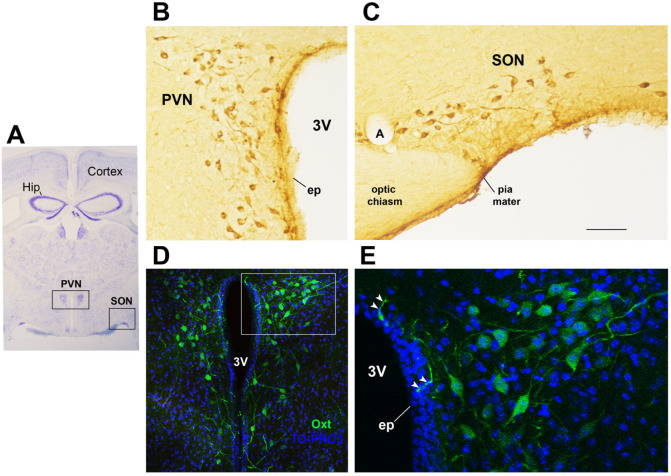
Oxytocinergic neurons in the paraventricular (PVN) and supraoptic (SON) hypothalamic nuclei. (**A**) Light microscopy (LM): Nissl-stained brain coronal section of the region corresponding to bregma −0.94 mm; PVN: hypothalamic paraventricular nucleus; SON: hypothalamic supraoptic nucleus; Hip: hippocampus. (**B**) LM: peroxidase immunohistochemistry of oxytocin (Oxt)-positive neurons in proximity of the ependymal layer (ep) of the third ventricle (3 V) in the PVN. (**C**) Peroxidase immunohistochemistry of Oxt neurons in the SON; A: artery. (**D**) Double-label confocal microscopy of Oxt neurons (green) and cell nuclei (blue TO-PRO3 staining) in the PVN. Panel (**E**) is an enlargement of the area framed in (**D**), showing Oxt-positive neurons and their projections (arrowheads) reaching and contacting the ependymal cells and the cerebrospinal fluid of the third ventricle (3 V). All figures refer to a 6-month-old male C57BL/6 mouse. Bregma reference sections from “The Mouse Brain Atlas”, Paxinos and Franklin (2001). The scale bar is included in C only and corresponds to different μm in each figure as follows: in (**A**): 1500 μm; in (**B**,**C**): 45 μm; in (**D**): 120 μm; in (**E**): 40 μm. All figures are original. Methodological details are available in the Appendix A.

**Figure 2 cells-11-00623-f002:**
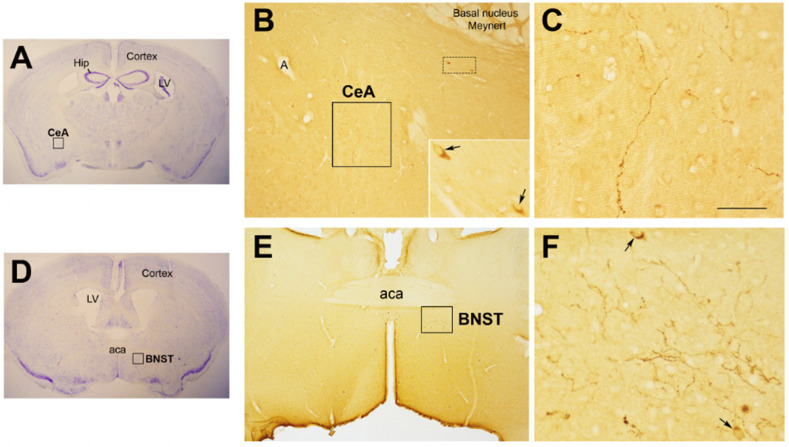
Oxytocinergic projections in the central nuclei of the amygdala (CeA) and in the bed nucleus of the stria terminalis (BNST). (**A**) Light microscopy (LM): Nissl-stained brain coronal section of bregma −0.94 mm; Hip: hippocampus; LV: lateral ventricle; CeA: central nuclei of the amygdala. (**B**) LM: peroxidase immunohistochemistry of oxytocin (Oxt), showing oxytocinergic fibers (framed area) and two parvocellular Oxt neurons (dotted framed area) in the CeA. Inset: enlargement of the dotted framed area, where Oxt-positive neurons are indicated by arrows. (**C**) Enlargement of the area framed in (**B**), rich in Oxt-positive fibers. (**D**) LM: Nissl-stained brain coronal section of the bregma 0.02 mm; BNST: bed nucleus of the stria terminalis; aca, anterior commissure. (**E**) LM: peroxidase immunohistochemistry of oxytocinergic projections in the BNST. (**F**) Enlargement of the area framed in (**E**) showing Oxt neurons (arrows) and Oxt projections in the BNST. All figures relate to a 6-month-old male C57BL/6 mouse. Bregma reference sections from “The Mouse Brain Atlas”, Paxinos and Franklin (2001). The scale bar is only specified in C and corresponds to different μm in each figure, as follows: in (**A**,**D**): 2300 μm; (**B**): 200 μm, inset 50 μm; in (**C**): 50 μm; in (**E**): 300 μm; in (**F**): 50 μm. All figures are original. Methodological details are available in the Appendix A.

**Figure 3 cells-11-00623-f003:**
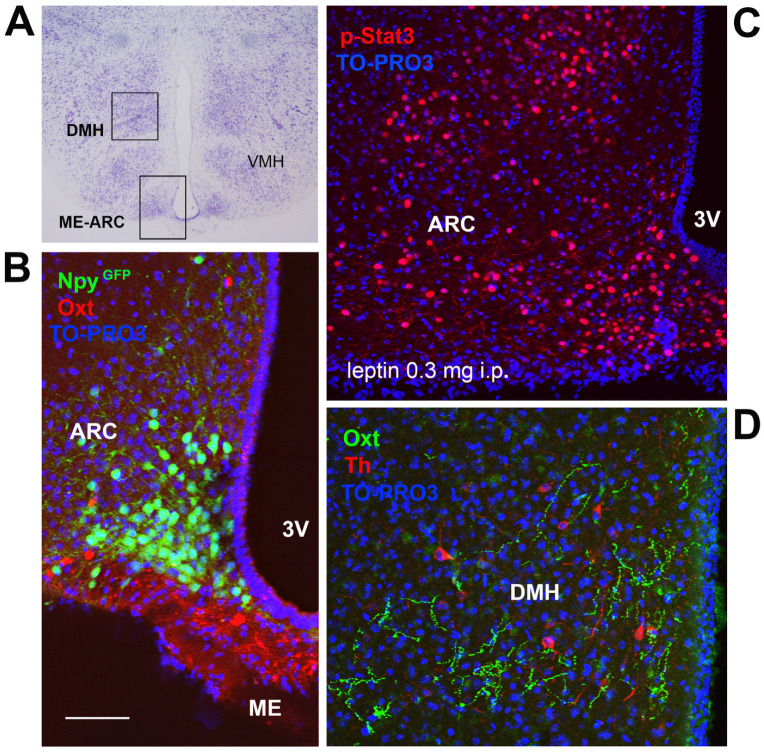
Hypothalamic nuclei regulating energy homeostasis. (**A**) Light microscopy: Nissl-stained brain coronal section of the region corresponding to bregma −3.13 mm, which includes the arcuate nucleus (ARC), median eminence (ME), dorsomedial hypothalamus (DMH), and ventromedial hypothalamus (VMH). (**B**) Confocal microscopy (CM) of oxytocin (Oxt) neurons and projections in close proximity of NPY neurons in the ARC-ME area from an NPY^Cre^; R26^GFP^ adult male mouse. (**C**) CM: p-Stat3-positive neurons in ARC-ME, 50 min following intraperitoneal (i.p.) injection of recombinant leptin (0.3 mg). p-Stat3 positivity indicates leptin-responsive neurons. (**D**) CM of DMH oxytocinergic projections; red staining indicates tyrosine hydroxylase (Th)-positive neurons. 3V: third ventricle; cell nuclei (blue) are marked using TO-PRO3 staining. Figure (**A**–**D**) refer to a 6-month-old male C57BL/6 mouse. Bregma reference sections from “The Mouse Brain Atlas”, Paxinos and Franklin (2001). The scale bar is only specified in C and corresponds to different μm in each figure, as follows: in (**A**): 400 μm; in (**B**): 50 μm; in (**C**): 65 μm; in (**D**): 40 μm. All figures are original. Methodological details are included in the Appendix A.

**Figure 4 cells-11-00623-f004:**
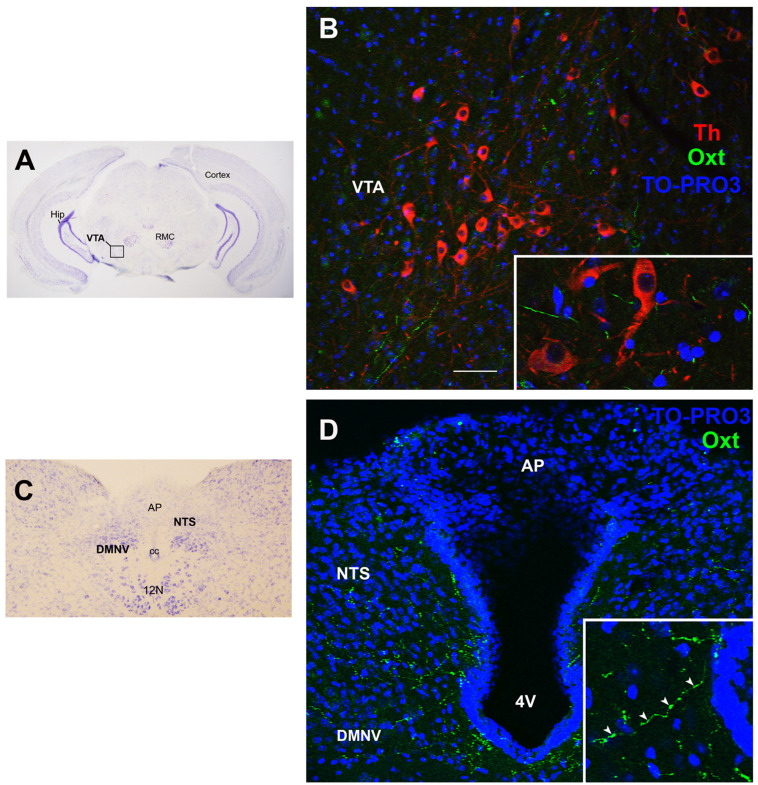
Oxytocin projections in the ventral tegmental area and in the dorsovagal complex. (**A**) Light microscopy (LM): Nissl-stained brain coronal section of the region corresponding to bregma −3.52 mm, where the ventral tegmental area (VTA; framed) is located, in proximity of the red magnocellular nuclei (RMC); Hip: hippocampus. (**B**) Triple-labeled confocal microscopy (CM) of oxytocinergic (Oxt) fibers scattered between dopaminergic tyrosine hydroxylase (Th)-positive VTA neurons. Inset: higher magnification of Th-positive neurons surrounded by Oxt fibers. (**C**) LM: Nissl-stained brain coronal section of the brainstem at bregma −7.32 mm showing the area postrema (AP), NTS, and dorsomotor nucleus of the vagus nerve (DMNV), which together form the dorsovagal complex (DMC), a critical structure for energy balance regulation; cc, central canal; 12N: hypoglossal nucleus. (**D**) Double-staining CM: Oxt projections in the DMC. Inset: Oxt fibers in the DMC at higher magnification. 4V: fourth ventricle. Cell nuclei (blue) are marked using TO-PRO3 staining. All figures refer to a 6-month-old male C57BL/6 mouse. Bregma reference sections from “The Mouse Brain Atlas”, Paxinos and Franklin (2001). The scale bar is only specified in (**C**) and corresponds to different μm in each figure, as follows: in (**A**): 2500 μm; in (**B**): 50 μm; inset 20 μm; in (**C**): 400 μm; in (**D**): 45 μm; inset 20 μm. All figures are original. Methodological details are available in the Appendix A.

**Figure 5 cells-11-00623-f005:**
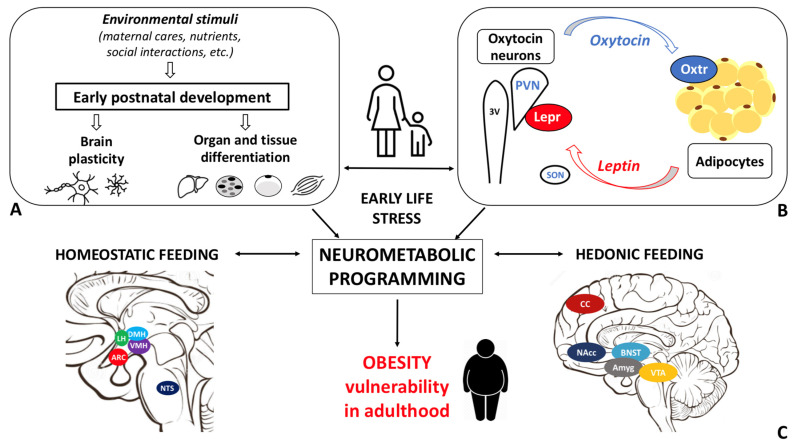
Schematic representations of the pathways linking early life stress to vulnerability to adult obesity. (**A**) Impact of environmental stimuli on the early postnatal development of various organs (e.g., brain, liver, brown and white adipose tissues, and skeletal muscle). (**B**) Oxytocin (Oxt), produced in the paraventricular nucleus (PVN) and supraoptic nucleus (SON) of the hypothalamus, regulates maternal bonding, stress response, and energy homeostasis. Oxt receptor (Oxtr) is highly expressed in white adipose tissue. Adipocytes are the main source of leptin (Lep), the master energy balance regulator, whose receptor (Lepr) is expressed in Oxt neurons. (**C**) Early adverse experiences may affect brain development, metabolically relevant organs differentiation, and the Oxt and Lep systems maturation. In this condition, dysfunctional neurometabolic programming and homeostatic and hedonic feeding centers maturation may occur, ultimately enhancing the vulnerability to obesity in adulthood. VMH: ventromedial, DMH: dorsomedial, LH: lateral hypothalamic nuclei; ARC: arcuate nucleus; NTS: nucleus of the solitary tract; CC: cerebral cortex; Nacc: nucleus accumbens; BNST: bed nucleus of the stria terminalis; Amyg: amygdala; VTA: ventral tegmental area.

**Table 1 cells-11-00623-t001:** Outstanding questions to be addressed.

1	Do the Oxt and Leptin systems influence each other’s development?
2	Is the Oxt–Lep systems interaction impacted by obesity?
3	Is Oxt the mediator of ELS-induced changes in metabolic health?
4	What are the consequences of ELS on short- and long-term eating behaviors, such as chow vs. palatable food consumption?
5	What are the consequences of ELS on total weight changes and metabolic health, such as adipose tissue development and resting energy expenditure?
6	What are the consequences of ELS on the vulnerability to an obesogenic environment in adulthood, such as adipose tissue dysfunction and metabolic abnormalities?

## Data Availability

Not applicable.

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
