# Peer review of "Early Life Stress, Brain Development, and Obesity Risk: Is Oxytocin the Missing Link?"

_cells, 2022, doi:10.3390/cells11040623_

Round 1

Reviewer 1 Report

The Review "Early life stress, brain development and obesity risk: is oxytocin the missing link?" by Colleluori aims to summarize the current knowledge on obesity, oxytocin, leptin and early life stress. The review is extremely well-written and presents its points in a clear and logical way. I have only very minor issues that I will list below:

The authors always included one scale bar per figure and gave different micrometer values in the figure legend. To my knowledge this is not common practice, normally each image contains its own scale bar. I am not asking to include a scale bar in each image, the method is equally valid, but to explain the procedure of only including one scale bar per figure in the figure legend.

The term for LG is "licking and grooming" and not "leaking and grooming".

To my knowledge "amygdala" is spelled with a "Y", not with an "i"

line 197: "The following opened questions..." should be "open"

line 219: "parallely" please remove "y"

Apart from those mentioned above I have no further issues, it was a pleasure to read and i can only highly recommend this well-written review for publication in "Cells"

Author Response

We thank the editor and the reviewers for their prompt and useful feedback, which we feel greatly enhanced the quality and clarity of our review. The text has been revised by a professional science translator. The point-by-point response (in black) to the comments of each reviewer (in blue) are reported below. Changes in the text are highlighted in yellow.

Reviewer 1

The Review "Early life stress, brain development and obesity risk: is oxytocin the missing link?" by Colleluori aims to summarize the current knowledge on obesity, oxytocin, leptin and early life stress. The review is extremely well-written and presents its points in a clear and logical way. I have only very minor issues that I will list below:

We are grateful to the reviewer for his/her positive comment. Changes in the text are highlighted in yellow.

The authors always included one scale bar per figure and gave different micrometer values in the figure legend. To my knowledge this is not common practice, normally each image contains its own scale bar. I am not asking to include a scale bar in each image, the method is equally valid, but to explain the procedure of only including one scale bar per figure in the figure legend.

Response: we agree that adding a description of the scale bar makes image interpretation easier and clearer. This information has been added to the text.

The term for LG is "licking and grooming" and not "leaking and grooming".

Response: thank you, the text has been amended.

To my knowledge "amygdala" is spelled with a "Y", not with an "i"

Response: thank you, the mistake has been corrected.

line 197: "The following opened questions..." should be "open"

Response: corrected. The section was moved to the end of the manuscript at the request of reviewer 4.

line 219: "parallely" please remove "y"

Response: corrected. The sentence has also been slightly modified to respond to a request by reviewer 3.

Apart from those mentioned above I have no further issues, it was a pleasure to read and i can only highly recommend this well-written review for publication in "Cells"

Thank you for your positive feedback!

Reviewer 2 Report

The authors performed a qualitative review of the role of oxytocin in obesity. Besides, the authors elaborated the impact of early life stress on obesity, by linking the oxytocin. The topic of this study is relevant. It is an interesting topic and there is a lack of knowledge about the role of oxytocin, particularly in human studies. There are, however, several areas in this manuscript worth addressing. Peripheral actions of oxytocin, particularly the myenteric and submucous ganglia, intestine, and liver are worth discussing. It would be helpful if the authors could discuss more details on the confounders of oxytocin, i.e., gender, other hormones, inflammation, in the discussion of the early life stress and oxytocin.

Author Response

We thank the editor and the reviewers for their prompt and useful feedback, which we feel greatly enhanced the quality and clarity of our review. The text has been revised by a professional science translator. The point-by-point response (in black) to the comments of each reviewer (in blue) are reported below. Changes in the text are highlighted in yellow.

Reviewer 2

The authors performed a qualitative review of the role of oxytocin in obesity. Besides, the authors elaborated the impact of early life stress on obesity, by linking the oxytocin. The topic of this study is relevant. It is an interesting topic and there is a lack of knowledge about the role of oxytocin, particularly in human studies.

We are grateful to the reviewer for this positive and useful feedback. The point-by-point response (in black) to each comment (in blue) is reported below. Changes in the text are highlighted in yellow.

There are, however, several areas in this manuscript worth addressing. Peripheral actions of oxytocin, particularly the myenteric and submucous ganglia, intestine, and liver are worth discussing.

Response: Thank you for this suggestion. Additional details regarding the peripheral oxytocin action have been added and briefly discussed. Please see lines 239-243 and 467-472.

It would be helpful if the authors could discuss more details on the confounders of oxytocin, i.e., gender, other hormones, inflammation, in the discussion of the early life stress and oxytocin.

Response: We thank the reviewer for this suggestion. We have added several sections (for example, see lines 188-194, 249-258, 308-310) to describe the sex-dependent differences in the response to ELS, to dietary habits, and in the oxytocin system. Furthermore, we now specify throughout the manuscript whether our findings referred to males or females. The relevance of the similarities between the Oxt and vasopressin systems are also reported as possible confounders when studying Oxt. The above changes have also been made to address the requests of reviewer 3.

Author Response

We thank the editor and the reviewers for their prompt and useful feedback, which we feel greatly enhanced the quality and clarity of our review. The text has been revised by a professional science translator. The point-by-point response (in black) to the comments of each reviewer (in blue) are reported below. Changes in the text are highlighted in yellow.

Reviewer 3

We are grateful to the reviewer for his/her prompt and useful feedback. The point-by-point response (in black) to each comment (in blue) is reported below. Changes in the text are highlighted in yellow.

  1. Describe literature search, how many hits where found with what search terms, define criteria for the exclusion and inclusion of the papers. Give dates for when was the search performed. There seems to be an arbitrary inclusion of selected papers for review with many gaps in the material selected.

Response: thank you. We clarified the scope of the paper (lines 96-98) and specified the type of evidence extracted from the literature. However, since our article is neither a meta-analysis nor a systematic review, and since such detailed information is not commonly provided in papers such as ours, we feel it may be unnecessary. We leave the decision as so its inclusion to the editor.

  1. Figures (1-3) have no references. If they are original (which is somewhat questionable for a review paper) than a supplement is needed with detailed M&M: in particular for the validation of the Oxytocin ab which is a neuropeptide composed of 9 amino acids, and its sequence and structure are very similar to arginine vasopressin. Thus the potential for cross reactivity is very high. The supplementary M&M should be detailed enough to be able to reproduce these findings: include the dissection and fixation protocols, tissue processing, the source of both primary and secondary abs and detection system as well as ab validating strategy.

Response: we agree with the reviewer and thank him/her for this useful comment. We have provided a detailed description of the methodologies used to obtain figures 1 to 4 in supplementary file 1. Antibody validation and cross-reactivity was performed by the manufacturers; additional details are provided in the product datasheets. Specifically, the antibodies recognize the complex Oxytocin-Neurophisin.

  1. Line: 19 should read “The prevalence of obesity” instead of Obesity

Response: Done. The sentence has been changed.

  1. Line:22 exertes should read “exerts”

Response: thank you.

  1. Line: 25-26 should read “on the vulnerability of obesity” and “on the role of oxytocin”

Response: The text has been amended. 

  1. Line: 27 worth should read “worthy”

Response: The text has been changed. 

  1. Line: 41-42 mention of hypertension, metabolic syndrome and cardiovascular diseases but there is no mention to the role of OT in ELS regarding these “most importantly” although a recent publication on the role of OT in ELS for specifically these very same indications has been published: McCook et al 2021 which focused not only on the heart and vasculature but also on the CNS and Vagal relationship (see also Iwasaki 2015). But in particular strengthens your thesis of the significance of OT in ELS mediated obesity.

Response: Thank you. We have added the article by McCook et al., 2021 for readers interested in further information on “ELS, Oxytocin and cardiovascular risk”, please see lines 516-517. Although the link between obesity and cardiovascular health is very strong, we do not discuss it in depth since it is outside the scope of our review.

  1. Line: 50 should read “the etiology of obesity”

Response: The text was changed as suggested. 

  1. Line 51: “Based on the foresight study in fact,” this statement makes no sense!

Response: the sentence was changed to improve its clarity.

  1. Line: 56 “converge to the dysfunction of the energy balance regulation.” this statement makes no sense!

Response: the sentence was changed to improve its clarity.

  1. Line: 58 should read, “The pathophysiology of obesity” Am no longer correcting the English the paper should be proofread by a native English speaker!

Response: We apologize, we had not realized that the references formatting software does not highlight most typos and mistakes. However, the manuscript has been revised by a professional science translator.

  1. Line: 61 please give specific examples of which organs and their dysfunction.

Response: Examples are now provided (lines 59-64).

  1. Line:65-66 please provide a reference.

Response: references were added to the text.

  1. Line:66-68 please provide a reference

Response: references were added to the text.

  1. Line: 91 please define ELS within this context with reference(s)

Response: a brief description of stress type is now provided in parentheses. A more detailed description of the ELS protocols selected for the review is now provided in the relevant section 2, (lines 110-114).

  1. Line: 92 please define inadequate maternal care

Response: “inadequate” has been changed to “disorganized or inconstant”. A more detailed description of the ELS protocols selected for the review is now provided in section 2 (lines 110-114).

  1. Line:151-152 it would be best if the references were placed after their respective reported association: weight gain or weight loss.

Response: we agree and have changed the text as suggested (lines 157-158).

  1. Line:218-220 “a finding revealed only in advanced vertebrates and thought to have developed parallelly (parallel) to species social and emotional behavioral complexity.” The statement makes no sense!

Response: we apologize. The sentence has been rephrased.

  1. Line: 274-277 should mention that these are clinical trials and should report their status and when completed their findings. Also, Zhang et al 2013 was the first to report a clinical trial on OT for obesity with beneficial results and should also be cited.

ClinicalTrials.gov Identifier: NCT02849743 Active, not recruiting

ClinicalTrials.gov Identifier: NCT03043053 Recruiting This is a randomized, double blind, placebo-controlled study of the effects of intranasal oxytocin in obese adults, ages 18-45 years old. Completed, no report.

ClinicalTrials.gov Identifier: NCT03119610 Completed This proof-of-concept study indicates that oxytocin may be useful for the treatment of sarcopenic obesity in older adults. Oxytocin administration may also provide additional cardiovascular benefits.

Response: thank you. The references formatting software does not recognize these entries. We have corrected the mistake and added the suggested reference (lines 754-760).

  1. Line:283-300 this section which is the crux of OT and ELS is highly deficient in studies on the subject: see McCook et al 2021, Ellis et al 2021, Wigger et al 2020, Krause 2018 etc but instead jumps to a description of Leptin which comes out of the bue!

Response: A separate section devoted to leptin has been added (please see paragraph 6). We have also included the studies by Ellis et al., 2021 and Krause et al., 2018 in the reference list. We refer the reader to the review by McCook et al., 2021 or Wigger et al., 2018 for more in-depth information on ELS-induced changes in the Oxt system as a mediator of the cardiovascular risk. Please see lines 328-332 and 516-517.

  1. Line 299-300: The differences between males and females is a very relevant topic to this perspective and it should be included as a separate section covering all points: ELS, oxytocin and leptin. The obesity affects men and women differently and that more women than men worldwide are obese should also be mentioned.

Response: we understand the relevance of this point. We have added several sections (for example, see lines 188-194, 249-258, 308-310) that describe the sex-dependent differences in the response to ELS, to dietary habits, and in the Oxt system. We now specify throughout the manuscript whether the findings refer to males or females.

  1. Line:301- 346 Leptin should have its own subsection. Furthermore, in contrast to the literature this section contains almost nothing on the role of ELS on Leptin. But it is in the section of ELS? It would be helpful for the reader to highlight similarities between the two systems in response to ELS. In other words, drawing parallels between what is known about oxytocin in ELS and leptin in ELS would make your hypothesis more tenable. For example, Shi et al 2020: show a relationship between the regulation of baroreflex measurements and leptin which is also reported for the oxytocin system as reviewed by McCook et al 2021. Another example would be the (line 298) “different spatio-temporal Oxtr expression patterns during development have been described in males and females” which is also described for the Leptin system (see below Ruigrock). Another parallel would be in what is reported in lines 289-292 and the similar finding by Shi et al 2020 for Leptin.

Response: we agree. Thank you for this useful input. We have added a separate paragraph (# 6) on leptin’s central action, its role in early brain development and how it is affected by ELS. However, given the length of our paper (which has now further increased) and the request of reviewer 4 to shorten it, we have not added an in depth discussion of the suggested parallelism between the Oxt and Lep systems. We leave this decision to the editor.

  1. Line: 318-320 Is instructive but neglects a very interesting and more recent study by Ramos-Lobo 2019 suggests: “We found that some defects previously considered irreversible due to neonatal deficiency of leptin signaling, including the poor development of arcuate nucleus neural projections, were recovered by LepR reactivation in adulthood. However, LepR deficiency in early life led to irreversible obesity via suppression of energy expenditure and to defects in the reproductive system and brain growth.” Ramos-Lobo2019

Response: We thank the reviewer for this useful input. The study is now discussed in the new section (paragraph 6). Please see lines 365-370.

Furthermore Ruigrock et al. 2021: report sex differences in nutrient sensing pathways and a targeted modulation of this pathway by ELS early in life. LepR sex differences although leptin levels did not differ and that sex differences in LepR expression seems to be region specific in the brain. Ruigrock 2021 and Bouillon-Minois2021 report on leptin after stress and that that response is also in a sex dependent manner.

Response: Thank you. We have discussed the finding of the meta-analysis by Bouillon-Minois et al., 2021 in the section devoted to leptin (paragraph 6). Please see lines 378-379.

  1. Line: 485-485 OxtR is not a gene! Furthermore the entire sentence makes no sense!

Response: We are not sure we understand the reviewer’s concern and hope that our response addresses it. We are available to make further changes. Youan et al. (2020) studied Oxtr expression in response to cold by RT-qPCR. The primers used for this purpose are detailed in their methodological section. Oxtr is listed as a gene in the NCBI database (https://www.ncbi.nlm.nih.gov/gene/?term=oxtr).The sentence has been changed to improve clarity.

  1. Line: 483 would have been a good place to discuss the interaction of OxtR and AVPR, and the complexity of the ligand receptor interaction. In ELS the receptor expression may be more important than circulating ligand levels see Wigger et al 2020, Krause et al 2018, McCook et al 2021, Ellis et al 2021.

Response: Thank you for this suggestion. We have added a brief sentence to emphasize the relevance of Oxtr as opposed to the ligand levels (lines 532-534). However, although we recognize its relevance, we have not discussed in depth Oxtr-Avpr interaction, which would open an additional topic. We now briefly mention it and refer the reader to other papers (lines 255-258). Since the length and contents of the review have already increased substantially, we have tried to find a compromise between the opposite requests of reviewers 3 and 4.

  1. Line: 493-494 Would be a good place to discuss the adverse effects of oxytocin administration in particular in ELS subjects: see McCook et al 2021 and Ellis et al 2021

Response: Thank you: differences in the response to Oxt administration based on ELS experiences are discussed in lines 328-332.

Reviewer 4 Report

Colleluori et al. present an interesting review on the links between early life stress, obesity and oxytocin. The work is relevant, comprehensive and generally well presented. Overall, I felt the manuscript can be shortened to improve conciseness and readability and the sections better articulated in a more cohesive argument. I have some other comments, which I detail below:

  1. Abstract: the abstract is too long on describing the rationale for the review and would benefit from including some take-home messages
  2. Before presenting and discussing specific topics, I think a brief introduction setting the research topic in context and highlighting why this review is important would be useful - this information is a bit lost throughout the manuscript.
  3. Are figure 1 - 4 original data or reproduced from somewhere else? This is not fully clear and needs to be clarified. I have nothing against the inclusion of original data in a review, but the reader might need to have access to the methods used to generate these data if the data have not been published before
  4. The authors do include a section on future directions; I think it would be nice if the readers could include a table with a list of 5-10 outstanding questions for future research; this might appeal to the reader and make the manuscript more compelling to future citations

Author Response

We thank the editor and the reviewers for their prompt and useful feedback, which we feel greatly enhanced the quality and clarity of our review. The text has been revised by a professional science translator. The point-by-point response (in black) to the comments of each reviewer (in blue) are reported below. Changes in the text are highlighted in yellow.

Reviewer 4

Colleluori et al. present an interesting review on the links between early life stress, obesity and oxytocin. The work is relevant, comprehensive, and generally well presented. Overall, I felt the manuscript can be shortened to improve conciseness and readability and the sections better articulated in a more cohesive argument.

Response: We are grateful to the reviewer for these positive comments. Several sections have been shortened and some concepts are now described more concisely. However, addressing the requests of the other reviewers has involved discussing additional notions, which have increased the length of the article. The point-by-point response (in black) to the comments of each reviewer (in blue) are reported below. Changes in the text are highlighted in yellow.

I have some other comments, which I detail below:

  1. Abstract: the abstract is too long on describing the rationale for the review and would benefit from including some take-home messages

Response: the abstract has been shortened from ~250 to ~200 words and a take home message has been added.

2. Before presenting and discussing specific topics, I think a brief introduction setting the research topic in context and highlighting why this review is important would be useful - this information is a bit lost throughout the manuscript.

Response: Thank you. Information concerning the research topic and its impact has been added at the end of paragraph 1. Please see lines 92-98.

  1. Are figure 1 - 4 original data or reproduced from somewhere else? This is not fully clear and needs to be clarified. I have nothing against the inclusion of original data in a review, but the reader might need to have access to the methods used to generate these data if the data have not been published before.

Response: the reviewer is right. We now specify that the images are original. A detailed description of the methodologies used to obtain them is reported in supplementary file 1.

4. The authors do include a section on future directions; I think it would be nice if the readers could include a table with a list of 5-10 outstanding questions for future research; this might appeal to the reader and make the manuscript more compelling to future citations

Response: Thank you for this interesting suggestion. We have added a brief section with some open questions for future research, which we agree makes the article more appealing. Please see Table 1.

Round 2

Reviewer 3 Report

The authors have addressed all my concerns. The revised manuscript is much improved and was a pleasure to read!

Reviewer 4 Report

The  authors have successfully addressed all of my points. I am delighted to recommend this manuscript for publication.